# Use of Antibiotics and Probiotics Reduces the Risk of Metachronous Gastric Cancer after Endoscopic Resection

**DOI:** 10.3390/biology10060455

**Published:** 2021-05-22

**Authors:** Junya Arai, Ryota Niikura, Yoku Hayakawa, Takuya Kawahara, Tetsuro Honda, Kenkei Hasatani, Naohiro Yoshida, Tsutomu Nishida, Tetsuya Sumiyoshi, Shu Kiyotoki, Takashi Ikeya, Masahiro Arai, Nobumi Suzuki, Yosuke Tsuji, Atsuo Yamada, Takashi Kawai, Kazuhiko Koike

**Affiliations:** 1Department of Gastroenterology, Graduate School of Medicine, The University of Tokyo, Tokyo 113-8655, Japan; jarai-tky@umin.ac.jp (J.A.); nsuzuki-ham@umin.ac.jp (N.S.); ytsuji-tky@umin.ac.jp (Y.T.); yamada-a@umin.ac.jp (A.Y.); kkoike-tky@umin.ac.jp (K.K.); 2Department of Gastroenterological Endoscopy, Tokyo Medical University, Shinjuku-ku, Tokyo 160-0023, Japan; t-kawai@tokyo-med.ac.jp; 3Clinical Research Promotion Center, The University of Tokyo Hospital, Tokyo 113-8655, Japan; tkawahara-tky@umin.ac.jp; 4Department of Gastroenterology, Nagasaki Harbor Medical Center, Nagasaki-shi, Nagasaki 850-8555, Japan; tetsupeacevo@yahoo.co.jp; 5Department of Gastroenterology, Fukui Prefectural Hospital, Fukui-shi, Fukui 910-0846, Japan; k-hasatani-nr@pref.fukui.lg.jp; 6Department of Gastroenterology, Ishikawa Prefectural Central Hospital, Kanazawa-shi, Ishikawa 920-8530, Japan; ynaohiro@ipch.jp; 7Department of Gastroenterology, Toyonaka Municipal Hospital, Toyonaka-shi, Osaka 560-8565, Japan; tnishida@gh.med.osaka-u.ac.jp; 8Department of Gastroenterology, Tonan Hospital, Sapporo-shi, Hokkaido 060-0004, Japan; t-sumiyoshi@tonan.gr.jp; 9Department of Gastroenterology, Shuto General Hospital, Yanai-shi, Yamaguchi 333-0801, Japan; shu2026eb@hi.enjoy.ne.jp; 10Department of Gastroenterology, St. Luke’s International Hospital, Chuo-ku, Tokyo 104-8560, Japan; takaike@luke.ac.jp; 11Department of Gastroenterology, Nerima Hikarigaoka Hospital, Nerima-ku, Tokyo 179-0072, Japan; araima-tky@umin.ac.jp

**Keywords:** metachronous gastric cancer, endoscopic resection, antibiotics, probiotics, gut microbiome

## Abstract

**Simple Summary:**

*Helicobacter pylori* is the most important cause of gastric cancer, and its eradication reduces the incidence of gastric cancer after endoscopic resection. However, incidence of metachronous gastric cancer is still high. More studies are needed to identify other chemopreventive drugs that may reduce the incidence of this disease. In this study, we focused on the alteration of the intragastric microbiome and examined the association between the use of antibiotics and probiotic drugs and risk of metachronous gastric cancer. Our findings suggest that the gut microbiome is associated with metachronous gastric cancer development.

**Abstract:**

Metachronous gastric cancer often occurs after endoscopic resection. Appropriate management, including chemoprevention, is required after the procedure. This study was performed to evaluate the association between medication use and the incidence of metachronous gastric cancer after endoscopic resection. This multicenter retrospective cohort study was conducted with data from nine hospital databases on patients who underwent endoscopic resection for gastric cancer between 2014 and 2019. The primary outcome was the incidence of metachronous gastric cancer. We evaluated the associations of metachronous gastric cancer occurrence with medication use and clinical factors. Hazard ratios were adjusted by age and Charlson comorbidity index scores, with and without consideration of sex, smoking status, and receipt of *Helicobacter pylori* eradication therapy during the study period. During a mean follow-up period of 2.55 years, 10.39% (140/1347) of all patients developed metachronous gastric cancer. The use of antibiotics other than those used for *H. pylori* eradication was associated with a lower incidence of metachronous gastric cancer than was non-use (adjusted hazard ratio (aHR) 0.56, 95% confidence interval (CI) 0.38–0.85, *p* = 0.006). Probiotic drug use was also associated with a lower incidence of metachronous gastric cancer compared with non-use (aHR 0.29, 95% CI 0.091–0.91, *p* = 0.034). In conclusion, the use of antibiotics and probiotic drugs was associated with a decreased risk of metachronous gastric cancer. These findings suggest that the gut microbiome is associated with metachronous gastric cancer development.

## 1. Introduction

*Helicobacter pylori* is the most important cause of gastric cancer [1], and its eradication reduces the incidence of gastric cancer after endoscopic resection [2]. However, long-term observational studies have shown a high incidence of metachronous gastric cancer, estimated at 2–3% per year, even after endoscopic resection and *H. pylori* eradication [3,4]. Although a randomized study conducted in Korea showed that *H. pylori* eradication reduced the risk of metachronous gastric cancer by approximately 50% [5], more studies are needed to identify other chemopreventive drugs that may reduce the incidence of this disease.

Several potentially chemopreventive drugs for gastric cancer have been identified. Statin use has been associated with a 10% reduced risk of gastric cancer [6], and the use of nonsteroidal anti-inflammatory drugs (NSAIDs) has been associated with a 6% reduction in gastric cancer incidence [7]. Recently, alteration of the intragastric microbiome in gastric cancer tissue has been studied [8]. Dysbiosis involving certain non–*H. pylori* bacteria may be associated with the gastric cancer risk [8]. Bacterial overgrowth may promote gastric carcinogenesis, even after *H. pylori* eradication [9]. Thus, we hypothesized that antibiotics and probiotic drugs might affect dysbiosis and reduce the risk of metachronous gastric cancer.

To address these issues, we performed a multicenter retrospective cohort study of patients who underwent endoscopic resection for gastric cancer and evaluated associations of the use of antibiotics, probiotic drugs, and other medications with the development of metachronous gastric cancer after endoscopic resection.

## 2. Materials and Methods

### 2.1. Study Design, Setting, and Participants

We performed this retrospective cohort study using combined diagnostic procedure databases from nine hospitals. The study period was April 2014–March 2019. The combined databases provided records for all inpatients at Tonan Hospital and for all inpatients and outpatients at the University of Tokyo Hospital, Shuto General Hospital, Fukui Prefectural Hospital, Nerima Hikarigaoka Hospital, St. Luke’s International Hospital, Toyonaka Municipal Hospital, Ishikawa Prefectural Central Hospital, and the Nagasaki Minato Medical Center. They contained data on diagnosis, comorbidities, and adverse events, recorded using the International Classification of Diseases, 10th revision (ICD-10), as well as on drugs and procedures, coded using the original Japanese system. We used the procedure codes for endoscopic resection, including endoscopic mucosal resection and endoscopic submucosal dissection (K6531–K6534), and the ICD-10 codes for gastric cancer (C160–166, C168, and C169), to extract data for all patients who underwent endoscopic resection for gastric cancer between April 2014 and March 2019. We excluded patients who underwent second endoscopic resections or additional surgical resections for gastric cancer <1 year after the index endoscopic resection. The follow-up period extended from the date of the first endoscopic resection to the final visit. The end of the follow-up period was March 2019, and loss to follow-up was defined using the date of the final visit. We also excluded the patients whose follow-up period was less than 1 year. This study was approved by the institutional review boards of the University of Tokyo Hospital (no. 2019161NI).

### 2.2. Outcomes and Variables

The primary outcome was the development of metachronous gastric cancer, as defined by ICD-10 codes (C160–C166, C168, and C169) and procedure codes for endoscopic and surgical resection >1 year after the index endoscopic resection (K6531–K6534, K654-2, K654-31, K654-32, K6551, K6552, K655-21, K655-22, K655-41, K655-42, K655-51, K655-52, K656, K656-2, K6571, K6572, K657-21, and K657-22).

We evaluated the following clinical factors: age, sex, smoking status, *H. pylori* eradication drug use, comorbidities, and medication use. Drug codes were used to assess *H. pylori* eradication, which was defined as the prescription of a 7-day course of clarithromycin- or metronidazole-based triple therapy for *H. pylori* infection between April 2014 and March 2019 (Appendix A). Age was dichotomized as >70 and ≤70 years. The following comorbidities (based on ICD-10 codes) were included: atrial fibrillation, acquired immunodeficiency syndrome, arterial thrombosis, carotid disease, cerebrovascular disease, chronic heart failure, chronic kidney disease (≤stage 5), dementia, diabetes mellitus with or without complications, deep vein thrombosis, hemiplegia, dyslipidemia, ischemic heart disease, liver disorder (mild/severe), malignancy with or without metastasis, pulmonary embolism, peripheral vascular disease, pulmonary disease, rheumatic disease, transient ischemic attack, peptic ulcer disease, unstable angina, and valvular disease. Charlson comorbidity index scores were calculated using these data [10]. Details of the ICD-10 codes are shown in Appendix A.

We assessed the use of antibiotics, probiotics, NSAIDs (including cyclooxygenase-2 (COX2) inhibitors), aspirin, metformin, statins with other lipid-lowering agents (e.g., fibrates), and proton pump inhibitors (PPI). The use of medications other than antibiotics was defined as a prescription for >60 days. Antibiotics were categorized into groups consisting of two or three major oral medications, such as aminoglycosides, carbapenems, penems, glycopeptides, cephems, tetracyclines, oxazolidinones, quinolones, lincomycin, chloramphenicol, metronidazole, penicillins, fosfomycin, macrolides, and sulfamethoxazole-trimethoprim combinations. Antibiotics used for *H. pylori* eradication therapy were not counted in antibiotic use. Antibiotics in the β-lactam group included penicillin, cephems, carbapenems, and penems. Antibiotics for anaerobic bacteria included lincomycin, metronidazole, carbapenem, and tetracyclines or combinations of β-lactamase inhibitors. Antibiotics for carcinogenic bacteria, such as Fusobacterium spp., Citrobacter spp., and Clostridium spp., included amoxicillin, fosfomycin, metronidazole, tetracycline, and lincomycin [11,12,13,14,15]. Probiotics included the combination of lactic acid bacteria, butyrate-producing bacteria, bifidobacteria, and saccharifying bacteria. Loxoprofen, sulindac, diclofenac, flurbiprofen, ibuprofen, indomethacin, ketoprofen, oxaprozin, naproxen, mefenamic acid, flufenamate aluminum, acemetacin, proglumetacin maleate, mofezolac, pranoprofen, tiaprofenic acid, zaltoprofen, tiaramide hydrochloride, etodolac, meloxicam, nabumetone, zaltoprofen, lornoxicam, and piroxicam, including COX2 inhibitors (celecoxib), were defined as NSAIDs. Statins included pitavastatin, simvastatin, pravastatin, fluvastatin, atorvastatin, and rosuvastatin. Fibrates included fenofibrate, bezafibrate, clinofibrate, and clofibrate. PPI included omeprazole, lansoprazole, rabeprazole, and esomeprazole. Details of the codes for these medications are shown in Appendix A.

### 2.3. Statistical Analysis

The primary endpoint, metachronous gastric cancer development, was censored on the date of the final visit. The Kaplan–Meier method was used to calculate the cumulative probability of metachronous gastric cancer development at 5 years. Univariate and multivariate Cox models were used to estimate hazard ratios and 95% confidence intervals (CIs). The multivariate Cox proportional hazards models were adjusted for age and Charlson comorbidity index scores, with and without consideration of sex and smoking status, as appropriate. *p* values < 0.05 were considered to be significant. All statistical analyses were performed using the SAS software (ver. 9.4; SAS Institute, Cary, NC, USA).

## 3. Results

### 3.1. Patient Characteristics

A total of 2832 patients underwent endoscopic resection for gastric cancer during the study period. After the exclusion of 1485 patients who underwent additional endoscopic or surgical re-section within 1 year and whose follow-up period was less than one year, data from 1347 patients were included in the analysis (Figure 1). The mean age of these 1347 patients was 72.54 years, and 75.87% (*n* = 1021) were male. Overall, 32.00% of the patients used antibiotics, 5.64% used probiotics, 7.20% used aspirin, 9.58% used NSAIDs, 1.41% used COX2 inhibitors, 10.02% used statins, 0.89% used fibrates, 1.04% used other lipid-lowering agents, 3.27% of patients used metformin, and 43.36% of patients used PPI. The baseline characteristics of the cohort are shown in Table 1. Antibiotics were principally used for infections, including respiratory tract infection (3.48%), urinary tract infection (3.02%), skin infection (0.70%), enteritis (0.70%), and biliary tract infection (0.93%).

### 3.2. Incidence of Metachronous Gastric Cancer after Endoscopic Resection and Associated Factors

Factors associated with metachronous gastric cancer, as determined by Cox modeling, are shown in Table 2 and Figure 2. In our patient sample, antibiotic use was associated with a lower incidence of metachronous gastric cancer compared with non-use (adjusted hazard ratio (aHR) 0.56, 95% CI 0.38–0.85, *p* = 0.006). The use of β-lactam antibiotics (aHR 0.61, 95% CI 0.39–0.96, *p* = 0.031), and amoxicillin, fosfomycin, metronidazole, tetracycline, or lincomycin (aHR 0.59, 95% CI 0.36–0.97, *p* = 0.038) was also associated with a decreased risk of metachronous gastric cancer. Probiotic use was associated with a lower incidence of metachronous gastric cancer compared with non-use (aHR 0.29, 95% CI 0.091–0.91, *p* = 0.034). Moreover, the use of statins was associated significantly with a lower incidence of metachronous gastric cancer (aHR 0.52, 95% CI 0.27–0.99, *p* = 0.047). Conversely, PPI use was associated with a higher risk of metachronous gastric cancer (aHR 1.49, 95% CI 1.05–2.11, *p* = 0.024). Clinical factors associated with a higher incidence of metachronous gastric cancer were age >70 years (aHR 1.57, 95% CI 1.09–2.28, *p* = 0.017) and peptic ulcer disease (aHR 2.95, 95% CI 2.05–4.24, *p* < 0.001; Appendix A). No interaction between antibiotics, probiotic drugs, or other medications was observed (Appendix A).

During the mean follow-up period of 2.55 years, 140 (10.39%) patients developed metachronous gastric cancer (of them, 29 patients eradicated *H. pylori* during the study period). The cumulative incidence of gastric cancer was 10.62% at 3 years and 12.48% at 5 years in patients using antibiotics, and 14.11% at 3 years and 35.68% at 5 years in patients not using antibiotics (Figure 3). Antibiotic use was associated significantly with a decreased incidence of gastric cancer (log-rank test, *p* = 0.026). This result was similar to another Kaplan–Meier analysis in the propensity score matched cohort, calculated by age, sex, CCI, and several infections (Appendix A). The cumulative incidence of gastric cancer was 6.06% at 3 and 5 years in patients using probiotics, and 12.95% at 3 years and 27.88% at 5 years in patients not using probiotics (Figure 3). Probiotic use was associated significantly with a decreased incidence of gastric cancer (log-rank test, *p* = 0.045).

## 4. Discussion

In this multicenter cohort study, we found that antibiotic and probiotic drug use was associated with a decreased risk of metachronous gastric cancer after endoscopic resection. In addition, statin use reduced the risk of metachronous gastric cancer development by 20%. Other medications did not significantly decrease this risk. These findings suggest that the gut microbiome is associated with metachronous gastric cancer development, and support previous findings of an association between dysbiosis and gastric cancer [8].

In this study, the use of all types of antibiotics was associated with a decreased incidence of metachronous gastric cancer (Figure 2). In particular, β-lactam antibiotic use decreased the risk of metachronous gastric cancer by 21%. Wide-spectrum antibiotics may affect various types of bacterium, consequently improving dysbiosis and decreasing the incidence of metachronous gastric cancer. Interestingly, the use of amoxicillin, fosfomycin, metronidazole, tetracycline, and lincomycin also tended to decrease the risk of metachronous gastric cancer. These antibiotics affect several bacterial strains that are reported to be carcinogenic in the gastrointestinal tract, including Fusobacterium spp., Citrobacter spp., and Clostridium spp. [11,12,13,14,15]. Our results are consistent with previous findings of associations between certain bacteria and gastric cancer. In our study, probiotics use was also associated significantly with a decreased incidence of metachronous gastric cancer (Figure 3B). Probiotics included butyrate-producing bacteria and Bifidobacterium, which have shown anti-carcinogenic effects in vitro and in vivo and may improve dysbiosis [16,17].

In contrast with previous studies [18,19,20], the associations of statin, aspirin, and COX2 inhibitor use with a decreased risk of metachronous gastric cancer did not reach statistical significance in this study. We suggest two reasons for this discrepancy. First, our study included patients with and without *H. pylori* infection, whereas previous studies included only patients in whom *H. pylori* had been eradicated. The difference in the prevalence of *H. pylori* infection might affect the chemopreventive potential of these drugs. The discrepancy may also be attributable to our examination of a relatively small number of patients over a short follow-up period, which may have limited our ability to detect the chemopreventive effects of these drugs.

Our study had several strengths. First, we evaluated associations of the use of a variety of medications with the incidence of metachronous gastric cancer after endoscopic resection. Second, the study had a multicenter cohort design. Nevertheless, our study had several limitations. First, it was retrospective. Second, our combined diagnostic procedure database lacked data on the success of *H. pylori* eradication before and during the study period. However, 5.75% of patients in our sample developed metachronous gastric cancer during the study period. In contrast, metachronous gastric cancer had developed in 3.30% of patients in an *H. pylori* eradication group and 8.82% of patients in a non-eradication group at the 3-year follow-up point in a previous study [2]. The incidence of metachronous gastric cancer in our study was consistent with that in the previous study. Thus, we speculate that the proportion of patients with *H. pylori* eradication did not differ significantly between our study and previous studies. Third, the follow-up period was relatively short and insufficient to accurately evaluate metachronous gastric cancer events. However, we will expand our database to double the number of hospitals and will perform further studies for another 2 years in the near future. Finally, the data was collected in a single country, Japan, where the incidence of gastric cancer is relatively high. The result of this study might not be applied to other countries where the incidence of gastric cancers is low as metachronous gastric cancer itself is very rare. Further studies in various countries are needed in the future.

## 5. Conclusions

In conclusion, the use of antibiotics and probiotics was associated with a decreased risk of metachronous gastric cancer. These medications may be candidate agents for the prevention of metachronous gastric cancer after endoscopic resection. These findings suggest that the gut microbiome is associated with metachronous gastric cancer development.

## Figures and Tables

**Figure 1 biology-10-00455-f001:**
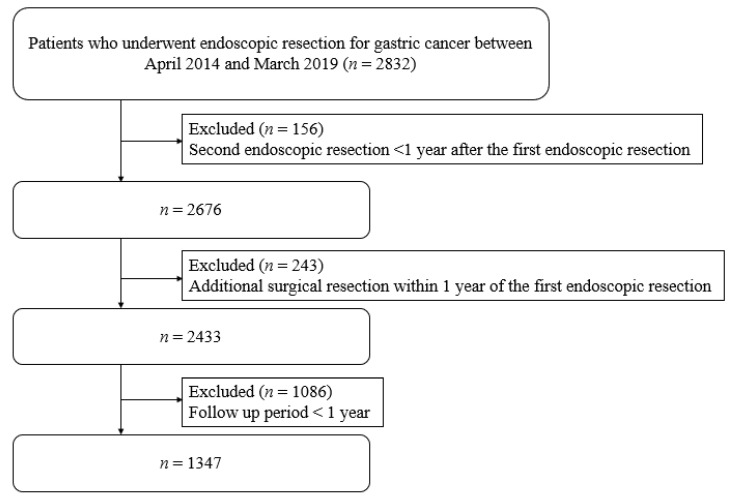
Flow of patient selection.

**Figure 2 biology-10-00455-f002:**
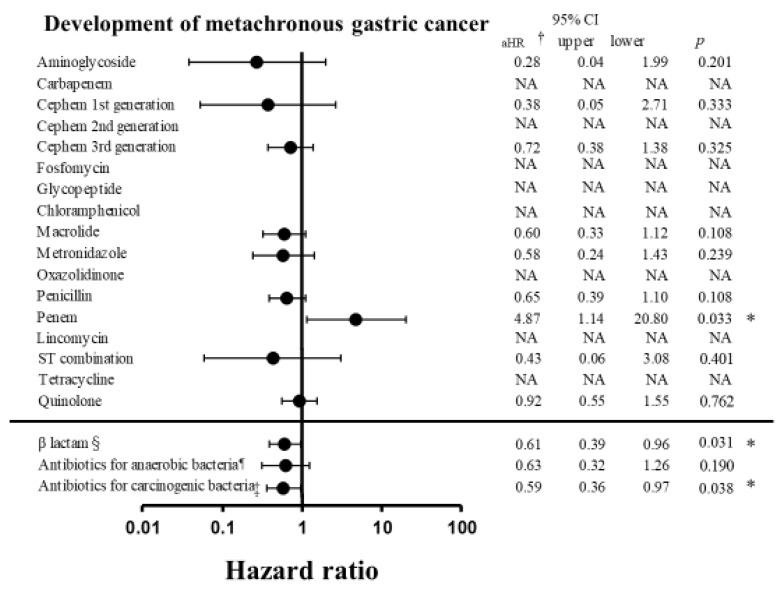
Associations of antibiotic use with metachronous gastric cancer development after endoscopic resection. Abbreviations: CI, confidence interval; aHR, adjusted hazard ratio; ST, sulfamethoxazole-trimethoprim. † HR adjusted for age >70 years and Charlson comorbidity index. § Antibiotics in the β-lactam class included penicillin, cephem, carbapenem, and penem. ¶ Antibiotics for anaerobic bacteria included lincomycin, metronidazole, carbapenem, tetracycline antibiotics, and combinations with β-lactamase inhibitors. ‡ Antibiotics for carcinogenic bacteria included amoxicillin, fosfomycin, metronidazole, tetracycline, and lincomycin. *: *p* values < 0.05 were considered to be significant.

**Figure 3 biology-10-00455-f003:**
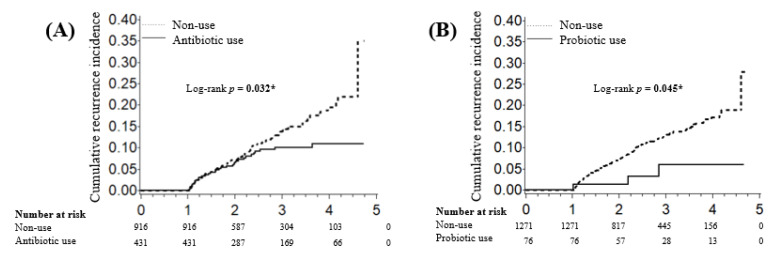
Cumulative incidence of metachronous gastric cancer in (**A**) antibiotic users vs. non-users and (**B**) probiotic users vs. non-users. Survival analysis was performed using the Kaplan–Meier method and log-rank test. * *p* values < 0.05 were considered to be significant.

**Table 1 biology-10-00455-t001:** Baseline characteristics of patients after endoscopic resection.

Characteristic	Number of Patients (%) or Mean ± SD
Sex (male)	1022 (75.87)
Age (>70 years)	845 (62.73)
Years of follow-up	2.56 ± 1.04
Smoking	685 (50.85)
*H. pylori* eradiation drug use during the study period	339 (25.17)
Charlson comorbidity index	1.52 ± 1.87
Medications	
Antibiotics	431 (32.00)
Aminoglycoside	21 (1.56)
Carbapenem	0 (0)
Cephem (first generation)	20 (1.48)
Cephem (second generation)	0 (0)
Cephem (third generation)	118 (8.76)
Fosfomycin	2 (0.15)
Glycopeptide	0 (0.00)
Chloramphenicol	0 (0)
Macrolide	159 (11.80)
Metronidazole	61 (4.53)
Oxazolidinone	0 (0)
Penicillin	200 (14.85)
Penem	4 (0.30)
Lincomycin	3 (0.22)
ST combination	18 (1.34)
Tetracycline	8 (0.59)
Quinolone	155 (11.51)
β-lactam §	301 (22.35)
Antibiotics for anaerobic bacteria ¶	108 (8.02)
Antibiotics for carcinogenic bacteria ‡	234 (17.37)
Probiotics	76 (5.64)
Aspirin	97 (7.20)
NSAIDs	129 (9.58)
COX2I (celecoxib only)	19 (1.41)
Statin	135 (10.02)
Fibrate	12 (0.89)
Other lipid-lowering agent	14 (1.04)
Metformin	44 (3.27)
Proton pump inhibitor	584 (43.36)

Abbreviations: SD, standard deviation; ST, sulfamethoxazole-trimethoprim; NSAID, nonsteroidal anti-inflammatory drug; COX2I, cyclooxygenase-2 inhibitor. § Antibiotics in the β-lactam class included penicillins, cephems, carbapenems, and penems. ¶ Antibiotics for anaerobic bacteria included lincomycin, metronidazole, carbapenem, and tetracycline antibiotics and combinations with β-lactamase inhibitors. ‡ Antibiotics for carcinogenic bacteria included amoxicillin, fosfomycin, metronidazole, tetracycline, and lincomycin.

**Table 2 biology-10-00455-t002:** Medications associated with metachronous gastric cancer after endoscopic resection.

Factor	Metachronous Gastric Cancer (*n* = 140)	No Metachronous Gastric Cancer (*n* = 1207)	Crude HR (95% CI)	Adjusted HR † (95% CI)	*p*
No antibiotic	105 (11.46)	811 (88.54)	1	1	
Antibiotics	35 (8.12)	396 (91.88)	0.66 (0.45–0.97)	0.56 (0.38–0.85)	0.006 *
No probiotic	137 (10.78)	1134 (89.22)	1	1	
Probiotics	3 (3.95)	73 (96.05)	0.33 (0.11–1.03)	0.29 (0.091–0.91)	0.034 *
No aspirin	128 (10.24)	1122 (89.76)	1	1	
Aspirin	12 (12.37)	85 (87.63)	1.08 (0.60–1.95)	0.91 (0.49–1.66)	0.747
No NSAID	127 (10.43)	1091 (89.57)	1	1	
NSAIDs	13 (10.08)	116 (89.92)	0.90 (0.51–1.59)	0.82 (0.46–1.47)	0.500
No COX2I	138 (10.39)	1190 (89.61)	1	1	
COX2I	2 (10.53)	17 (89.47)	0.81 (0.20–3.27)	0.85 (0.21–3.43)	0.814
No statin	130 (10.73)	1082 (89.27)	1	1	
Statins	10 (7.41)	125 (92.59)	0.60 (0.31–1.14)	0.52 (0.27–0.99)	0.047 *
No fibrate	138 (10.34)	1197 (89.66)	1	1	
Fibrates	2 (16.67)	17 (83.33)	1.61 (0.40–6.48)	1.56 (0.39–6.29)	0.535
No other lipid-lowering agent	140 (10.50)	1193 (89.50)	1	1	
Other lipid-lowering agents	0 (0.00)	14 (100.00)	NA	NA	-
No metformin	135 (10.36)	1168 (89.64)	1	1	
Metformin	5 (11.36)	39 (88.64)	0.95 (0.39–2.32)	0.81 (0.32–2.03)	0.647
No PPI	59 (7.73)	704 (92.27)	1	1	
PPI	81 (13.87)	503 (86.13)	1.57 (1.12–2.198)	1.49 (1.05–2.11)	0.024 *

Abbreviations: HR, hazard ratio; CI, confidence interval; NSAID, nonsteroidal anti-inflammatory drug; COX2I, cyclooxygenase-2 inhibitor; PPI, proton pump inhibitor. † HR adjusted for age >70 years, sex, smoking status, and Charlson comorbidity index. *: *p* values < 0.05 were considered to be significant.

## Data Availability

All data presented in this study are included within the paper and its supplementary files.

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
