# Peer review of "Use of Antibiotics and Probiotics Reduces the Risk of Metachronous Gastric Cancer after Endoscopic Resection"

_biology, 2021, doi:10.3390/biology10060455_

Round 1

Reviewer 1 Report

1) It is well known that metachronus recurrence occurs in constant rate over about 10 years after ESD.  Therefore, this kind of study whose primary endpoint was metachronous gastric cancer development needs long-term follow-up  longer than at least 3 years.  In this retrospective study, however, a mean follow-up period was only 1.55 years. Authors can extend the follow-up duration through adjusting the study period (for example 2014-2019 --> 2014-2016).

2) Authors argue that  the use of antibiotics and probiotic drugs was associate with a decreased risk of metachronous gastric cancer. To support this conclusion more robustly and focus on main findings, propensiy matched analysis would be better than current study design. As study population is quite large (n=2433), I think propensiy matched analysis can be feasible.

3) In this study, H. pylori eradication was defined as the prescription of 7-day course of triple therapy, not the real eradication after treatment. It is confusing and authors need to change the terminology.

4) As authors did not know the treatment results, multivariate analysis results in this study were not adjusted for real eradication results of H. pylori after treatment, which can lead to significant bias.

Author Response

1) It is well known that metachronus recurrence occurs in constant rate over about 10 years after ESD.  Therefore, this kind of study whose primary endpoint was metachronous gastric cancer development needs long-term follow-up  longer than at least 3 years.  In this retrospective study, however, a mean follow-up period was only 1.55 years. Authors can extend the follow-up duration through adjusting the study period (for example 2014-2019 --> 2014-2016).

Response: Thank you for this comment. We excluded the patients whose follow-up period was less than one year to extend the follow-up period. As a result, the mean follow-up period was 2.55 years, just a little less than 3 years in the revised manuscript. However, our main results did not change in the revised manuscript. We have added this exclusion criteria, and changed results (line 97-98, and results section).

2) Authors argue that the use of antibiotics and probiotic drugs was associate with a decreased risk of metachronous gastric cancer. To support this conclusion more robustly and focus on main findings, propensiy matched analysis would be better than current study design. As study population is quite large (n=2433), I think propensiy matched analysis can be feasible.

Response: We have estimated propensity score using factors of antibiotics use, calculated by age, sex, Charlson comorbidity index, and various infections, and performed a 1:1 propensity score matched analysis. These results were consistent with those of the all cohort. We have added these results in supplementary figure 4.

3) In this study, H. pylori eradication was defined as the prescription of 7-day course of triple therapy, not the real eradication after treatment. It is confusing and authors need to change the terminology.

Response: We agree with your comment. We have changed the word “eradication of H, pylori” to “H. pylori eradication drug use”.

4) As authors did not know the treatment results, multivariate analysis results in this study were not adjusted for real eradication results of H. pylori after treatment, which can lead to significant bias.

Response: We agree with your comment. We excluded the factor of H. pylori eradication therapy from the adjusting factors for multivariable analysis. In addition, we have added the sentence about lacked data on the success of H. pylori eradication in the limitations.

Reviewer 2 Report

The study :Use of antibiotics and probiotics reduces the risk of metachronous gastric cancer after endoscopic resection by  Junya Arai

Is a good one study on a wide group of endoscopicaly resected patients in a multicentric experience.  The study is well presented and documented , moreover is pleasable to read.

I have not great suggestions .

-My only request is to understand if possible why patients took antibiotics; if this isn’t a totally known data, it is possible to insert some example of the most frequent indications.

-Have  authors any ideas about fig 2 Penem antibiotics? Few patients  are considered ,but this  is the only antibiotic very different from the others. Were they used to cure a particular disease?

-The simple summary present conclusions more intriguing than the abstract ones. Une of the 2 las sentence may be contracted adding  suggestion on gut microbiome.

-Gut microbiome may be added in Key words

Author Response

The study :Use of antibiotics and probiotics reduces the risk of metachronous gastric cancer after endoscopic resection by  Junya Arai

Is a good one study on a wide group of endoscopicaly resected patients in a multicentric experience.  The study is well presented and documented , moreover is pleasable to read.

I have not great suggestions .

-My only request is to understand if possible why patients took antibiotics; if this isn’t a totally known data, it is possible to insert some example of the most frequent indications.

Response: Thank you for your suggestion. Antibiotics were principally used for infections including respiratory tract infection (3.48%), urinary tract infection (3.02%), skin infection (0.70%), enteritis (0.70%), and biliary tract infection (0.93%). We have added this sentence in result paragraph (lines 165-167).

-Have authors any ideas about fig 2 Penem antibiotics? Few patients  are considered ,but this  is the only antibiotic very different from the others. Were they used to cure a particular disease?

Response: No. Penem antibiotics in fig2 was not carbapenem. That was a hybrid structure of penicillin and cephalosporin antibiotics. The indication of penem antibiotics was respiratory tract infection (25%) and urinary tract infection (75%) in our study.

-The simple summary present conclusions more intriguing than the abstract ones. Une of the 2 las sentence may be contracted adding suggestion on gut microbiome.

Response: Thank you. We have added the sentence as follows: "In conclusion, the use of antibiotics and probiotic drugs were associated with a decreased risk of metachronous gastric cancer. These findings suggest that the gut microbiome may be associated with metachronous gastric cancer development." (Lines 52-53, and 264-265)

-Gut microbiome may be added in Key words

Response: Thank you for this comment. I agree with your opinion. I added “gut microbiota” in key word.

Reviewer 3 Report

The study under review is an interesting retrospective multicenter cohort study on risk and preventive factors of metachronous gastric cancer after endoscopic resection. 

The manuscript is well-written and the study objective is of scientific interest and patient-relevant. The title reflects properly the subject of the paper. The abstract provides an accessible summary of the paper. The key messages are short, accurate, and clear. The methods used are appropriate and the data support the conclusions.

There are no major flaws, no major revisions are required. I have only one question to be answered: what were the indications for all the different types of antibiotics and probiotics in all these patients??? Did they actually receive these medications for prevention of metachronous cancer or what else were the concrete indications? This should be explained in detail in the manuscript before the study under review can be considered for publication in Cancers. Thanks!

Author Response

The study under review is an interesting retrospective multicenter cohort study on risk and preventive factors of metachronous gastric cancer after endoscopic resection.

The manuscript is well-written and the study objective is of scientific interest and patient-relevant. The title reflects properly the subject of the paper. The abstract provides an accessible summary of the paper. The key messages are short, accurate, and clear. The methods used are appropriate and the data support the conclusions.

There are no major flaws, no major revisions are required. I have only one question to be answered: what were the indications for all the different types of antibiotics and probiotics in all these patients??? Did they actually receive these medications for prevention of metachronous cancer or what else were the concrete indications? This should be explained in detail in the manuscript before the study under review can be considered for publication in Cancers. Thanks!

Response: Thank you for this comment. No, patients did not receive medications for prevention of metachronous gastric cancer. Antibiotics were principally used for infections including respiratory tract infection (3.48%), urinary tract infection (3.02%), skin infection (0.70%), enteritis (0.70%), and biliary tract infection (0.93%). We added this sentence in result paragraph (lines 165-167).